# D-Garment: Physically Grounded Latent Diffusion for Dynamic Garment Deformations

**Antoine Dumoulin**                                     *antoine.dumoulin@inria.fr*
*Inria Centre at the University Grenoble Alpes*

**Adnane Boukhayma**                                     *adnane.boukhayma@inria.fr*
*Inria, University of Rennes, CNRS, IRISA-UMR 6074*

**Laurence Boissieux**                                   *laurence.boissieux@inria.fr*
*Inria Centre at the University Grenoble Alpes*

**Bharath Bhushan Damodaran**                            *bharath.damodaran@interdigital.com*
*InterDigital Inc.*

**Pierre Hellier**                                       *pierre.hellier@inria.fr*
*Inria, University of Rennes, CNRS, IRISA-UMR 6074*

**Stefanie Wuhrer**                                      *stefanie.wuhrer@inria.fr*
*Inria Centre at the University Grenoble Alpes*

**Reviewed on OpenReview:** *https://openreview.net/forum?id=NrPyio1aUK*

## Abstract

We present a method to dynamically deform 3D garments, in the form of a 3D polygon mesh, based on body shape, motion, and physical cloth material properties. Considering physical cloth properties allows to learn a physically grounded model, with the advantage of being more accurate in terms of physically inspired metrics such as strain or curvature. Existing work studies pose-dependent garment modeling to generate garment deformations from example data, and possibly data-driven dynamic cloth simulation to generate realistic garments in motion. We propose *D-Garment*, a learning-based approach trained on new data generated with a physics-based simulator. Compared to prior work, our 3D generative model learns garment deformations conditioned by physical material properties, which allows to model loose cloth geometry, especially for large deformations and dynamic wrinkles driven by body motion. Furthermore, the model can be efficiently fitted to observations captured using vision sensors such as 3D point clouds. We leverage the capability of diffusion models to learn flexible and powerful generative priors by modeling the 3D garment in a 2D parameter space independently from the mesh resolution. This representation allows to learn a template-specific latent diffusion model. This allows to condition global and local geometry with body and cloth material information. We quantitatively and qualitatively evaluate *D-Garment* on both simulations and data captured with a multi-view acquisition platform. Compared to recent baselines, our method is more realistic and accurate in terms of shape similarity and physical validity metrics. Code and data are available for research purposes at `https://dumoulina.github.io/d-garment/`.

## 1 Introduction

Dressing avatars with dynamic garments is a long-standing challenge in computer graphics and learning-based modeling. Garments are used in virtual applications, ranging from entertainment industries such as

video games and animation, to fashion with clothing design and virtual try-on. In applications involving an avatar such as telepresence and virtual change rooms, an important use case is accurately fitting a dynamic garment model to observations captured using vision sensors such as 2D videos, or dynamic 3D point clouds.

We consider the problem of deforming a 3D garment based on physical inputs, via learning a conditional generative model of 3D geometry deformation. Given a garment template mesh, physical parameters of the cloth material, and the wearer's body shape, pose and preceding motion, our method models the distribution over dynamic garment geometries $p(\text{garment deformation} \mid \text{body motion}, \text{shape}, \text{material})$, and outputs temporally consistent 3D deformations.

Existing work can be categorized into two main classes. Pose-dependent garment models (Vidaurre et al., 2024; De Luigi et al., 2023; Zheng et al., 2024; Shen et al., 2020; Corona et al., 2021) output a garment draped over a body given a characterization of the cloth, a body shape, and a pose. These methods have been successfully tested on downstream tasks such as garment reconstruction from observations and garment retargeting to different users. However, the problem is under-constrained as the 3D geometry of the same garment on the same body performing the same pose can vary depending on the velocity and acceleration of the body parts. Existing work outputs a motion-independent solution, which prevents these methods from generating temporal garment details caused by different dynamics.

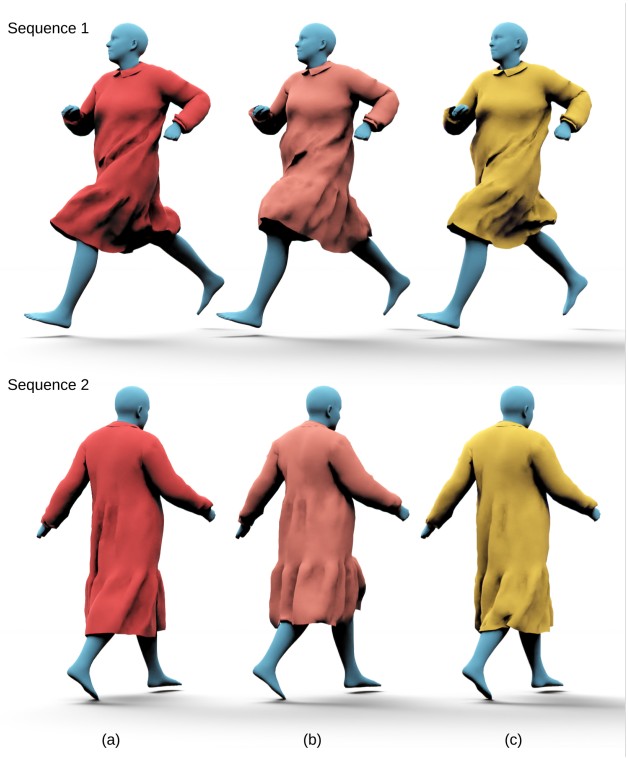

Sequence 1

Sequence 2

(a)      (b)      (c)

Figure 1: For two different motions (lines) and three varying cloth materials (colors), *D-Garment* uses a latent diffusion model to generate dynamic garment deformations from physical inputs defined by a cloth material and the underlying body shape and motion. Our model can represent large deformations and fine wrinkles.

In contrast, dynamic cloth simulation methods (Bertiche et al., 2022; Tiwari et al., 2023; Santesteban et al., 2022; Chen et al., 2024; Grigorev et al., 2023) realistically animate dynamic clothes on humans in motion. They produce physically plausible deformations given a garment in rest-pose, body shapes and motions. However, these models leverage cloth motion descriptors to predict the next cloth state. Hence, they are not designed for fitting to observations captured using vision sensors.

To allow for dynamic garments that can be fitted to observations using optimization, we present a physically grounded conditional generative model of 3D garment deformation. We define cloth material as the combination of the three physical quantities bending, stretching and density. To work at high resolution, we take advantage of a latent diffusion model (Ho et al., 2020; Rombach et al., 2022) in a two-dimensional *uv*-space to conditionally deform an arbitrary fixed garment template. Hence, our model is template-specific and needs to be trained per garment template. The model provides fine-grained control over input conditions related to the person wearing the garment and the garment's material. The diffusion model defines a distribution over plausible garment deformations, enabling the generation of multiple consistent outputs conditioned on the same input.

To efficiently condition our model on motion and body shape information, we leverage a parametric body model and condition the garment deformation on body shape and a sequence of poses describing the motion. While our model is agnostic to the parametric body model as long as it decouples identity and pose *e.g.* Neophytou & Hilton (2013); Pishchulin et al. (2017); Loper et al. (2015); Xu et al. (2020); Mihajlovic et al.

(2022), we use SMPL (Loper et al., 2015) as it is easy to integrate. For physical grounding, we condition our model on parameters controlling cloth material by training the model with the output of a physics-inspired simulator. While any physics-inspired simulator that allows controlling material via parameters is applicable *e.g.* Baraff & Witkin (1998); Provot (1995); Narain et al. (2012); Bouaziz et al. (2014); Li et al. (2018), we use a projective dynamics simulator (Ly et al., 2020) due to computational efficiency and a low number of parameters.

For training and evaluation, we use the physics-based simulator to generate a synthetic dataset of a dynamically deforming wide dress with complex geometry. The dress is simulated using different body shapes, motions, and material parameters. The resulting dataset contains 172 unique body motions from AMASS (Mahmood et al., 2019). Each motion is randomly performed by 3 body shapes and simulated with 3 cloth materials. The sewing pattern of the dress used for simulation was chosen to correspond to a dress in 4DHumanOutfit (Armando et al., 2023), a dataset of reconstructed dressed 3D humans in motion acquired using a multi-view camera platform. This allows for evaluation on acquired data.

We apply our model to a registration task, where we optimize our model's parameters to fit the result to dynamic 3D point cloud sequence of 4DHumanOutfit. We build a two stage pipeline that first fits a parametric body model to the sequence, and subsequently optimizes the latent vector of our model by minimizing the Chamfer distance.

Comparative experiments show that our approach outperforms state-of-the-art baselines in terms of geometric shape fidelity and physically inspired metrics due to its capability of disentangling body motion and cloth material.

In summary, the main contributions of this work are:

- A physically grounded method to dynamically deform a fixed 3D garment template to match input cloth materials, body shape and motion, based on a 2D latent diffusion model.

- A dataset, available at `https://doi.org/10.57745/GZTNJC`, of a dynamic 3D dress with complex sewing pattern simulated for different body shapes, motions and cloth materials. It contains simulations of 172 motions with variations of body shape and cloth material, totaling over 1500 sequences.

- A registration method fitting our model's parameters to dynamic point cloud sequences of captured dressed humans in motion.

## 2 Related Work

Dynamic garment modeling is a growing research area in computer vision. Current learning-based models can be categorized into two classes: pose-dependent garment modeling and dynamic cloth simulation. Achar et al. (2024) provide an exhaustive review on cloth draping methods. We focus on works that learn garment models, which in particular excludes works that reconstruct possibly animatable garments.

**Pose-Dependent Garment Modeling.** Modeling 3D garments is traditionally time-consuming and requires expertise. Data-driven cloth models alleviate this process by enabling parametric 3D modeling and automatic cloth-body draping. These models can be applied to various tasks, such as 3D model reconstruction from images or point clouds. SMPLicit (Corona et al., 2021) is a parametric model capable of generating diverse garments with a fine-grained control over cloth cut and design. Follow-up works modeled diverse garments given different inputs such as image (Sarafianos et al., 2025), 2D masks (Zheng et al., 2024), sketches (Wang et al., 2018) and more recently text (He et al., 2024; Nakayama et al., 2025; Bian et al., 2025). To fit the generated 3D model over different body shapes, a parametric body model (Loper et al., 2015) is commonly used. The garment can be posed using the body's Linear Blend Skinning (LBS) weights. This straightforward approach needs to compute LBS weights of the garment w.r.t. the body and does not model pose-dependent high-frequency details.

Table 1: Positioning of our work w.r.t. existing learning-based 3D garment models that generalize over multiple body shapes and poses. Our model additionally generalizes across materials, models dynamic details, and allows fitting to observations in the form of dynamic 3D point clouds.

| | Generalization across cloth material | Models dynamic details | Allows fitting to observations |
|---|:---:|:---:|:---:|
| **Pose-Dependent Garment Modeling** | | | |
| TailorNet (Patel et al., 2020) | | | |
| DiffusedWrinkles (Vidaurre et al., 2024) | | | ✓ |
| Laczkó et al. (2024) | | | |
| Cape (Ma et al., 2020) | | | ✓ |
| Shen et al. (2020) | ✓ | | ✓ |
| DrapeNet (De Luigi et al., 2023) | | | ✓ |
| SkiRT (Ma et al., 2022) | | | ✓ |
| Shi et al. (2024) | | ✓ | |
| **Dynamic Cloth Simulation** | | | |
| Santesteban et al. (2021) | | ✓ | |
| SNUG (Santesteban et al., 2022) | | ✓ | |
| GAPS (Chen et al., 2024) | | ✓ | |
| HOOD (Grigorev et al., 2023) | ✓ | ✓ | |
| ContourCraft (Grigorev et al., 2024) | ✓ | ✓ | |
| MGDDG (Zhang et al., 2022) | | ✓ | ✓ |
| Ours | ✓ | ✓ | ✓ |

Yang et al. (2018) studied cloth deformation across different body poses and motions by analyzing statistics of clothing layer deformations modeled w.r.t. an underlying parametric body model. While this allows to generalize to new factors, the model has limited capacity.

TailorNet (Patel et al., 2020) proposed a scalable model that generalizes across body poses by predicting low and high frequency wrinkles driven by body shape, pose and garment style. DiffusedWrinkles (Vidaurre et al., 2024) leveraged a diffusion model in $uv$-space to generate wrinkles conditioned like TailorNet. Another line of works (Ma et al., 2020; Laczkó et al., 2024; Shen et al., 2020) models deformations directly over the body surface learning pose-dependent information. DrapeNet (De Luigi et al., 2023) splits the problem into a 3D generative model and a draping model. A recent approach based on point clouds (Ma et al., 2022) learns dynamic LBS weights to model humans in loose clothing but it is not adapted for generative modeling. Shi et al. (2024) built a transformer model capable of synthesizing dynamic garments from body sequences.

Most of these approaches represent static deformations driven by a single pose, omitting dynamic deformations driven by body motion and the physical properties of the garment. Loose garments pose significant challenges due to reliance on either pre-computed LBS weights or deformations defined on the body surface. In contrast, we propose a latent diffusion model conditioned on physics-informed parameters, body pose and motion, enabling the generation of temporally coherent dynamic garments. Our approach is capable of capturing both static and dynamic wrinkles which are traditionally predicted by cloth simulators.

**Dynamic Cloth Simulation.** Physics-based simulation is a traditional approach in computer graphics for garment animation (Nealen et al., 2006; Baraff & Witkin, 1998). While state-of-the-art simulators are capable of accurately reproducing cloth dynamics (Li et al., 2018; Romero et al., 2021), these methods are computationally expensive. Learning-based simulation has been introduced to accelerate computation (Pfaff et al., 2021; Sanchez-Gonzalez et al., 2020; Kairanda et al., 2024). The line of work most closely related to ours focuses on cloth-body interactions to animate dynamic garments over humans (Li et al., 2024; Santesteban et al., 2022; Chen et al., 2024; Grigorev et al., 2023; 2024; Zhang et al., 2022). The use of a parametric body model, instead of a generic rigid-body mesh, allows to predict dynamic deformations from a canonical space leveraging LBS for pose and shape deformations.

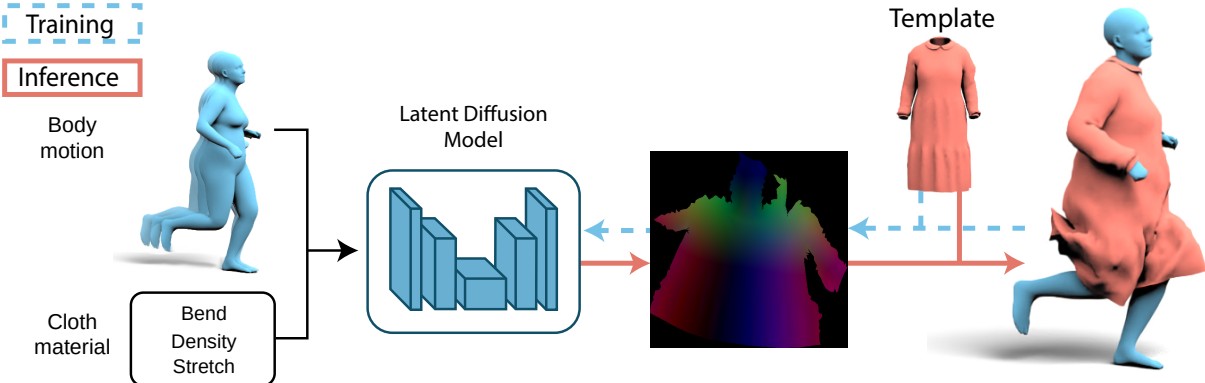

Figure 2: *D-Garment* generates garment deformations conditioned on body shape, motion and cloth material. It builds upon a 2D latent diffusion model (Sec. 3.2) to learn how to deform a template in *uv*-space (Sec. 3.1). 3D mesh vertex displacement from template is parameterized by the *uv* displacement map, and our model is trained on it along with the conditional inputs. At inference, our model generates the deformed garment by iteratively denoising the Gaussian noise w.r.t. its conditional inputs.

Recent works focus on deforming garments in response to body motion. Santesteban et al. (2021) formulated the motion given a current pose and body part velocities and accelerations. SNUG (Santesteban et al., 2022) introduced a self-supervised model to prune computationally expensive data generation. GAPS (Chen et al., 2024) improved SNUG by predicting LBS weights adapted to loose garments. HOOD (Grigorev et al., 2023) built a graph neural network enabling free-flowing motion without relying on LBS for posing. ContourCraft (Grigorev et al., 2024) added new losses to HOOD reducing cloth-body collisions and cloth self-penetration. Motion Guided Deep Dynamic Garments (MGDDG) (Zhang et al., 2022) predicted garment deformations in a generative way given previous states of garment and body.

Note that to achieve consistent cloth dynamics, current learned-based cloth simulation methods leverage previous cloth states either explicitly with vertices velocity (Grigorev et al., 2023; 2024; Zhang et al., 2022) or implicitly using recurrent architecture (Santesteban et al., 2021; 2022; Chen et al., 2024). In contrast, our approach is not auto-regressive, thereby enabling dynamic garment deformations at any given frame facilitating tasks such as fitting to observations.

**Positioning.** Table 1 positions our work w.r.t. scalable 3D garment models that generalize over body shape and pose. Our work combines advantages of existing works and allows for generalization across materials, dynamic detail modeling, and fitting to observations.

## 3 Method

Given a garment template represented as 3D triangle mesh $\mathcal{T}$, our goal is to learn a conditional generative model $\mathcal{G}$, called *D-Garment*, that deforms $\mathcal{T}$ into new mesh instances $\mathcal{M}$ while keeping the mesh topology fixed. We condition the generation on the underlying body shape, pose, pose velocity, and physical material properties of the garment. Conditioning on physical material properties allows to physically ground the model. At test time, *D-Garment* can be used for generation and fitting.

*D-Garment* represents a dynamic garment on top of a parametric human body model that decouples body shape and pose parameters. In our implementation, we use SMPL (Loper et al., 2015), and represent body shape $\boldsymbol{\beta}$ and pose sequence $\boldsymbol{\theta}_{t-2}, \boldsymbol{\theta}_{t-1}, \boldsymbol{\theta}_t$ as concatenation of the two preceding poses and the current one, where $t$ denotes a discrete time step and $l = 2$ the pose sequence length. The representation of cloth material is inspired by traditional works in physics-inspired simulation (Baraff & Witkin, 1998) and includes stretch coefficient $\mathbf{s}$ (in $N/m$), mass density coefficient $\mathbf{d}$ (in $kg/m^2$) and bending coefficient $\mathbf{b}$ (in $N \cdot m$) as $\boldsymbol{\gamma} := [\mathbf{s}, \mathbf{d}, \mathbf{b}]$. Parameter $\mathbf{s}$ controls resistance to stretching or compression, $\mathbf{d}$ controls the influence

of inertia, and **b** controls resistance to bending or curvature changes. These parameters will be used to physically ground the model.

Inspired by the recent success of diffusion models in 2D domain generation, our 3D mesh generator $\mathcal{G}$ consists of a 2D latent diffusion model, based on the latent diffusion architecture of Rombach et al. (2022). Our model, illustrated in Figure 2, directly generates a mesh deformation

$$\mathcal{M}_t \sim \mathcal{G}(\boldsymbol{\theta}_{t-2}, \boldsymbol{\theta}_{t-1}, \boldsymbol{\theta}_t, \boldsymbol{\beta}, \boldsymbol{\gamma}). \tag{1}$$

This formulation is independent of intermediate body-driven skinning, unlike prior works *e.g.* (Patel et al., 2020; Zhang et al., 2022; De Luigi et al., 2023; Vidaurre et al., 2024), which removes the need to rig $\mathcal{T}$.

### 3.1 Garment Representation

To leverage powerful diffusion models in the 2D domain for generation, we reframe our 3D generation problem by representing garment deformations in a 2D domain encoding $(u, v)$ coordinates. Inspired by *geometry images* from Gu et al. (2002), we encode relative 3D mesh vertex displacements from $\mathcal{T}$ into a 2D geometric displacement map $D$ via a pre-computed parametrization $\phi : \mathbb{R}^2 \to \{1, 2, \ldots, n\}$ where $n$ is the number of vertices of $\mathcal{T}$.

Given an inferred displacement map $\hat{D}$, the inverse $uv$-map $\phi^{-1}$ allows to lift pixels to mesh vertices as

$$\hat{\mathcal{M}}^k = \mathcal{T}^k + \hat{D}^{\phi^{-1}(k)}, \tag{2}$$

where $\hat{\mathcal{M}}$ is the resulting mesh and $k$ is the vertex index. Inversely, forward mapping, combined with triangle barycentric coordinate-based interpolation, computes a continuous displacement map for a deformed mesh $\mathcal{M}$ relatively to its template $\mathcal{T}$ for a given 2D coordinate $(u, v)$ as

$$D^{(u,v)} = \mathcal{M}^{\phi(u,v)} - \mathcal{T}^{\phi(u,v)}. \tag{3}$$

### 3.2 Diffusion Model

Using a canonical 2D representation of 3D garment meshes allows to benefit from well established scalable 2D latent diffusion architectures and their pre-trained weights. Being the state-of-the-art in text to image generation, diffusion models have been extended to various applications (Po et al., 2024). The latent diffusion model introduced by Rombach et al. (2022) is made of a variational auto-encoder (VAE) that maps high resolution images to a lower resolution latent space where reverse diffusion is learned based on denoising diffusion probabilistic models (Ho et al., 2020) by a denoising network $\epsilon_\theta$. The key strength of diffusion models lies in their reversibility. The forward process involves learning to predict artificially added noise in images. The reverse process, formulated as an iterative refinement, enables the model to gradually remove noise from the latent space, reconstructing high-quality data step by step. We adapt this model to learn generation of geometric displacement maps with conditions that allow for physical grounding.

**Training.** First, we adapt the VAE of Podell et al. (2024) to our mesh displacement map data distribution $\{\mathcal{M}_t, D_t\}_t$. Since the statistics of this displacement map differs from natural images, pre-trained VAE would be sub-optimal. We finetune the decoder via an extended training loss combining $L_2$ reconstruction error $||\hat{D} - D||_2$ and the displaced vertex-to-vertex error $||\hat{\mathcal{M}} - \mathcal{M}||_2$. Next, the conditional denoiser $\epsilon_\theta$ is trained with denoising score matching using samples $\{D_t, \boldsymbol{\theta}_{t-2}, \boldsymbol{\theta}_{t-1}, \boldsymbol{\theta}_t, \boldsymbol{\beta}, \boldsymbol{\gamma}\}_t$ from our training data corpus as

$$\mathbb{E}_{\mathbf{z}, \epsilon, s} \left[ ||\epsilon - \epsilon_\theta(\mathbf{z}_s, s | \boldsymbol{\theta}_{t-2}, \boldsymbol{\theta}_{t-1}, \boldsymbol{\theta}_t, \boldsymbol{\beta}, \boldsymbol{\gamma})||_2^2 \right], \tag{4}$$

where $\mathbf{z} \in \mathbb{R}^{64 \times 64 \times 4}$ is the latent for $D_t$, $\epsilon \sim \mathcal{N}(0, I)$, $s$ the diffusion time step, and the noisy latent obtained with forward diffusion $\mathbf{z}_s = \alpha_s \mathbf{z} + \sigma_s \epsilon$. The scaling factor and standard deviation of the forward diffusion are

$$\bar{\alpha}_s = \prod_{u=1}^{s}(1 - \beta_u), \quad \alpha_s = \sqrt{\bar{\alpha}_s}, \quad \sigma_s = \sqrt{1 - \bar{\alpha}_s}, \tag{5}$$

where variance $\beta_u$ is determined by the noise schedule.

**Inference.** At test time, a latent Gaussian noise $\mathbf{z}_T$ is iteratively denoised with $\epsilon_\theta$ to generate a latent displacement map $\mathbf{z}_0$, conditioned on body shape, poses and cloth material. The VAE then decodes $\mathbf{z}_0$ into a displacement map $\hat{D}$, which allows to compute mesh $\hat{\mathcal{M}}$ with Equation (2). Several algorithms can be used to reverse the latent diffusion (*e.g.* DDPM (Ho et al., 2020) or DDIM (Song et al., 2021)) with varying levels of inference speed, sample quality and stochasticity. We use the DPM-Solver++ (Lu et al., 2022) sampler thanks to its quality for the number of denoising steps required.

**Fitting.** Inspired by a registration method (Guo et al., 2024), we use our model to deform $\mathcal{T}$ to fit observations represented as a 3D point cloud sequence $\mathcal{P}$. As $\mathcal{P}$ typically contains additional points belonging to the body or other garments, we fit our model to $\mathcal{P}$ using Chamfer Distance $\mathcal{L}_{\mathrm{CD}}$ with the adaptive robust function $\rho$ (Barron, 2019), a cloth-body collision loss $\mathcal{L}_c$ and a regularisation loss $\mathcal{L}_{reg}$ consisting of Laplacian smoothing (Nealen et al., 2006), normal consistency and edge length minimization. This can be viewed as generating sample meshes $\mathcal{M}^*$ that match $\mathcal{P}$, leading to the optimization of the latent $\mathbf{z}_T^*$ as:

$$\mathbf{z}_T^* = \arg \min_{\mathbf{z}_T} \rho\left(\mathcal{L}_{\mathrm{CD}}(\mathcal{P}, \mathcal{M}^*)\right) + \mathcal{L}_c + \mathcal{L}_{reg}, \tag{6}$$

where $\mathcal{M}^*$ is generated in a differentiable way through our sampler from a latent $\mathbf{z}_T^*$, conditioned on $\boldsymbol{\gamma}$ and the body motion $\boldsymbol{\beta}, \boldsymbol{\theta}_i$'s. The latent $\mathbf{z}_T^*$ is optimized with Adam (Kingma & Ba, 2014).

Inspired by the clothing energy term in Yang et al. (2016), $\mathcal{L}_c$ is defined as:

$$\mathcal{L}_c = \sum_{v \in \mathcal{M}^*} \delta_{\mathrm{in}}(v, \mathcal{B}) \min_{b \in \mathcal{B}} ||v - b||_2, \tag{7}$$

where $\delta_{\mathrm{in}}$ is an indicator function for points inside a mesh, and $\mathcal{B}$ contains the body mesh vertices.

## 4 Dataset

Prior works often use datasets containing tight garments, with little dynamic effects. To test our approach, we created a dataset of a simulated 3D loose dress. We use a design similar to one of the outfits in 4DHumanOutfit (Armando et al., 2023), allowing to test our model on multi-view reconstructions of a real dress. The dress is simulated over 172 motion sequences, performing walking and running motions from AMASS (Mahmood et al., 2019). For each sequence, three body shapes were uniformly sampled following $\boldsymbol{\beta} \sim \mathcal{U}(-1, 1)$, where $\mathcal{U}$ is a uniform distribution. Furthermore, for each sequence, three cloth materials (bending **b**, stretching **s**, density **d**) were also uniformly sampled following $\boldsymbol{\gamma} \sim \mathcal{U}([10^{-8}, 10^{-4}], [40, 200], [0.01, 0.7])$. Each sequence was simulated for all combinations of the sampled parameters.

**Simulation.** We used a simulator based on projective dynamics (Ly et al., 2020) which is less physically accurate but 10 times faster than implicit solvers. It requires a single value for each cloth material parameter: bending, stretching, density. Other simulation parameters, which do not only depend on the cloth (friction, air damping, collision tolerance), were fixed across all sequences. The simulation frame rate was 50 fps. To initialize the simulation, the garment size was manually draped over the canonical body model, avoiding intersections. The garment was simulated over 100 frames, linearly interpolating from the zero pose and shape to the first frame of the sequence. It was then stabilized for 100 frames to reduce noisy motion.

**Data cleaning.** SMPL can contain self-intersections, which cause unsolvable cloth-body intersections for the simulator. To solve this, we pre-processed AMASS using an optimization used in Tzionas et al. (2016) to minimize self-penetration. We optimized for body joints that are the most disruptive in simulation *i.e.* arms and shoulders. Angular velocity of poses and the distance to the initial poses were also minimized to regularize the output and maintain temporal consistency. We post-processed simulated cloth sequences with remaining cloth-body intersections by pushing vertices outside following the nearest body normal. Some simulation failures have been manually removed from the dataset.

# 5 Experiments

**Implementation details.** We implemented the inverse $uv$ mapping $\phi^{-1}$ using bi-linear interpolation. We found that the interpolation technique (*e.g.* nearest, bi-linear or bi-cubic interpolation) is marginally impacting the mesh quality thanks to our high resolution image (512x512). By averaging texture coordinates that correspond to a single mesh vertex, we did not notice any discontinuity on the surface geometry across seams. To avoid remaining collision artifacts, the vertices are pushed outside the body mesh in a simple post-processing step ablated in Table 3.

We used a pretrained VAE from SDXL (Podell et al., 2024) downscaling the images to latents as $\mathbb{R}^{512 \times 512 \times 3} \rightarrow \mathbb{R}^{64 \times 64 \times 4}$. The decoder part was finetuned on $40k$ steps and then frozen during $\epsilon_\theta$ training. The denoiser network $\epsilon_\theta$ was built with an U-net (Ronneberger et al., 2015) that takes the encoded latents to 5 convolutional layers of output channel size $(64, 128, 256, 512, 512)$ and conditions via cross-attention (Vaswani et al., 2017). The following conditional input dimensions were used: $\boldsymbol{\gamma} \in \mathbb{R}^3$, $\boldsymbol{\beta} \in \mathbb{R}^8$, $\boldsymbol{\theta} \in \mathbb{R}^{111}$ using 6 dimensional rotations (Zhou et al., 2019) of 18 body joints and a global translation. Both decoder and $\epsilon_\theta$ weights were optimized using AdamW optimizer (Loshchilov & Hutter, 2019). We have trained $\epsilon_\theta$ on 50 epochs using a batch size of 32 lasting 10 days on a single NVidia A6000. The noising schedule was set to a cosine variance (Lin et al., 2024) increasing from $\beta_1 = 10^{-4}$ to $\beta_T = 0.02$ sampled in 1000 diffusion steps during training and 20 steps during inference using SDE-DPM++ (Lu et al., 2022). The inference achieves an interactive frame rate of 7.5 fps. We have used the Diffusers (von Platen et al., 2022) library to implement the diffusion model and PyTorch3D (Ravi et al., 2020) for the regularization loss $\mathcal{L}_{reg}$.

## 5.1 Evaluation Protocol

**Dataset.** The garment geometry and body poses are normalized according to the current body global rotation and translation for each frame. The template $\mathcal{T}$ is the mean geometry of the normalized dataset. We compute its $uv$ parametrization using OptCuts (Li et al., 2018) which jointly minimizes the distortion and the seam lengths, limiting under-sampling and discontinuities induced by seams.

We take 3 motions from the training dataset with corresponding variations of body shape and cloth material and change each factor at a time to an unseen value. For the body motion factor we use 3 unseen motions, namely "*C19-runtohoptowalk*", "*B16-walkturnchangedirection*" and "*B17-walktohoptowalk1*". This results in 3 test sets for unseen materials, body motions and body shapes, which allows to assess how well models generalize for each factor independently. We also build a challenging test set with all combinations of unseen factors. We randomly leave out 5% of the remaining training data for evaluation purposes.

**Metrics.** We quantitatively compare the geometric shape fidelity of the results to the test set with common 3D computer vision metrics. Vertex error $E_v$ shows the accuracy between the prediction and the ground truth. Chamfer Euclidean distance $E_{CD}$ assesses the shape similarity of the prediction and the ground truth. Chamfer normal distance $E_n$ allows to compare the wrinkling between the prediction and the ground truth. Nearest vertices corresponding normals are compared using absolute cosine similarity. In the following, we denote by $U$ the set of uniformly sampled points over the mesh surface $\mathcal{M}$.

$$E_v = \frac{1}{|\mathcal{M}|} \sum_{v \in \mathcal{M}} ||v - \hat{v}||, \qquad E_{CD} = \frac{1}{|U|} \sum_{v \in U} \min_{\hat{v} \in \hat{U}} ||\hat{v} - v|| + \frac{1}{|\hat{U}|} \sum_{\hat{v} \in \hat{U}} \min_{v \in U} ||\hat{v} - v||,$$

These shape similarity metrics allow to estimate how well a method reproduces deformations compared to the simulator. However, they fail to evaluate physical quality. To address this limitation, we propose several physically inspired metrics inspired by Santesteban et al. (2022). The cloth-body collision percentage $E_c$ measures the amount of clothing predicted inside the body. Higher values indicate physically implausible clothing or simulation failure. Wrinkling error $E_b$, related to bending $\mathbf{b}$, compares dihedral angles to the ground truth. Area strain error $E_s$, related to stretching $\mathbf{s}$, compares the in-plane deformation from the rest shape to the ground truth. The center of mass distance $E_d$, related to material density $\mathbf{d}$, compares the global rigid motion to the ground truth. Note that $E_b$, $E_s$ and $E_d$ are independent of each other and provide

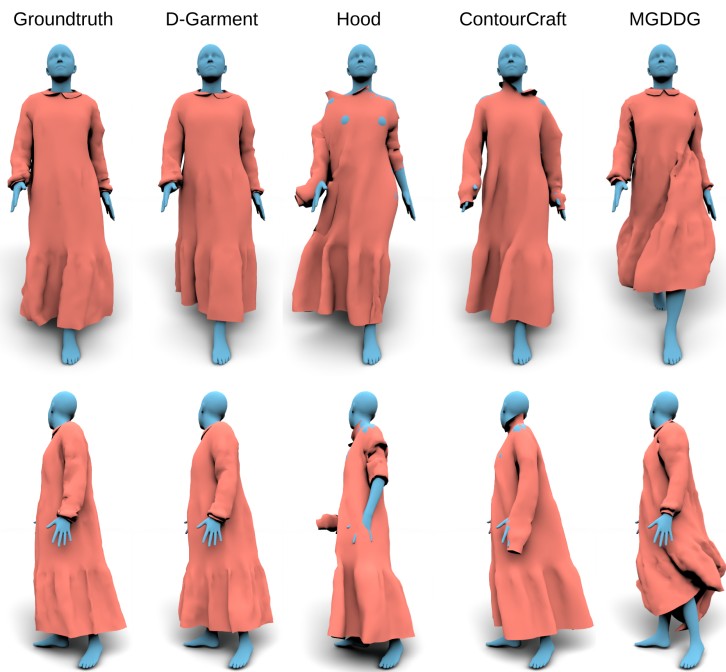

Figure 3: We qualitatively compare *D-Garment* to HOOD (Grigorev et al., 2023), ContourCraft (Grigorev et al., 2024) and MGDDG (Zhang et al., 2022). *D-Garment* generates visually plausible wrinkles without requiring cloth velocity or position input. Corresponding video can be viewed in the supplementary material.

Table 2: Quantitative comparison to HOOD (Grigorev et al., 2023), ContourCraft (Grigorev et al., 2024) and MGDDG (Zhang et al., 2022). Best scores are in **bold**. Time indicates the inference time.

| | Unseen body shape | | | | | | | Unseen motion | | | | | | | Time |
| | Shape Similarity | | | Physical Validity | | | | Shape Similarity | | | Physical Validity | | | | |
| | $E_v \downarrow$ | $E_{CD} \downarrow$ | $E_n \downarrow$ | $E_c \downarrow$ | $E_b \downarrow$ | $E_s \downarrow$ | $E_d \downarrow$ | $E_v \downarrow$ | $E_{CD} \downarrow$ | $E_n \downarrow$ | $E_c \downarrow$ | $E_b \downarrow$ | $E_s \downarrow$ | $E_d \downarrow$ | |
|---|---|---|---|---|---|---|---|---|---|---|---|---|---|---|---|
| HOOD | 14.10 | 1.49 | 0.59 | 1.69 | 0.80 | 4.81 | 9.46 | 21.21 | 2.37 | 0.65 | 1.67 | 0.88 | 4.71 | 16.68 | 0.08s |
| Cont.Craft | 9.69 | 0.43 | 0.53 | 1.43 | 0.72 | 6.31 | 5.03 | 11.10 | 0.48 | 0.56 | 1.25 | 0.74 | **4.42** | 7.13 | 0.23s |
| MGDDG | 8.31 | 0.33 | 0.56 | 1.10 | 0.81 | 27.60 | 4.02 | 9.53 | 0.38 | 0.59 | 1.13 | 0.84 | 36.51 | 4.95 | 0.35s |
| *D-Garment* | **3.49** | **0.10** | **0.41** | **0.54** | **0.60** | **3.94** | **1.51** | **6.14** | **0.25** | **0.48** | **0.70** | **0.64** | 5.00 | **2.88** | 0.13s |

| | Unseen material | | | | | | | Unseen all factors | | | | | | |
| | Shape Similarity | | | Physical Validity | | | | Shape Similarity | | | Physical Validity | | | |
| | $E_v$ | $E_{CD}$ | $E_n$ | $E_c$ | $E_b$ | $E_s$ | $E_d$ | $E_v$ | $E_{CD}$ | $E_n$ | $E_c$ | $E_b$ | $E_s$ | $E_d$ |
|---|---|---|---|---|---|---|---|---|---|---|---|---|---|---|
| HOOD | 25.31 | 3.24 | 0.67 | 1.58 | 0.91 | **4.61** | 20.96 | 19.75 | 2.52 | 0.61 | 1.79 | 0.82 | 7.65 | 14.88 |
| Cont.Craft | 10.28 | 0.46 | 0.57 | 1.30 | 0.73 | 5.62 | 5.68 | 9.75 | 0.38 | 0.53 | 1.08 | 0.70 | 7.59 | 6.34 |
| MGDDG | 8.68 | 0.34 | 0.58 | 1.00 | 0.84 | 29.08 | 3.77 | 8.51 | 0.34 | 0.56 | 0.96 | 0.81 | 31.23 | 4.28 |
| *D-Garment* | **3.41** | **0.10** | **0.42** | **0.43** | **0.59** | 5.08 | **1.64** | **4.94** | **0.19** | **0.45** | **0.72** | **0.59** | **7.02** | **2.34** |

a measure of isolated characteristics. They are defined as

$$E_c = \frac{100}{|U|} \sum_{v \in U} \delta_{\text{in}}(v, \mathcal{B}), \qquad E_b = \sqrt{\frac{1}{|E|} \sum_{e \in E} (\Theta(e) - \Theta(\hat{e}))^2}, \qquad E_s = |\Psi_s(\mathcal{M}) - \Psi_s(\hat{\mathcal{M}})|,$$

where $E$ is the set of edges of mesh $\mathcal{M}$, $\Theta(e)$ is the signed dihedral angle between adjacent faces, and $\Psi_s(\mathcal{M})$ is the integral area strain deformation from the rest pose.

## 5.2 Comparative Evaluation to Baselines

We compare *D-Garment* to methods that allow for generalization across cloth materials and that model dynamic details. While one of the first approaches allowing for varying motions falls in this category (Yang

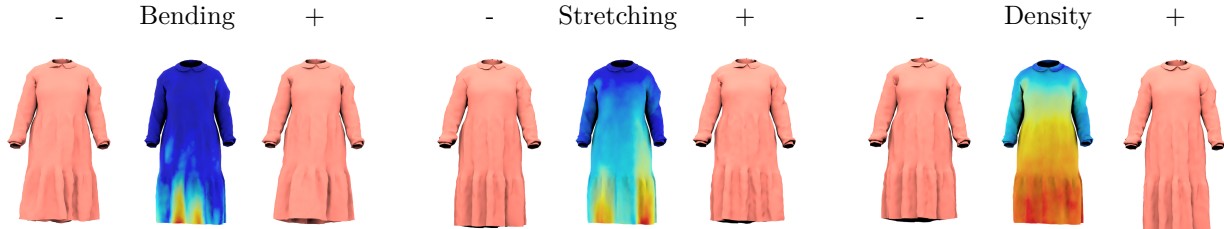

Figure 4: Varying one cloth material parameter at a time. For each parameter, the color map shows per-vertex distances between minimal and maximal values. 0 ████████ 10 cm

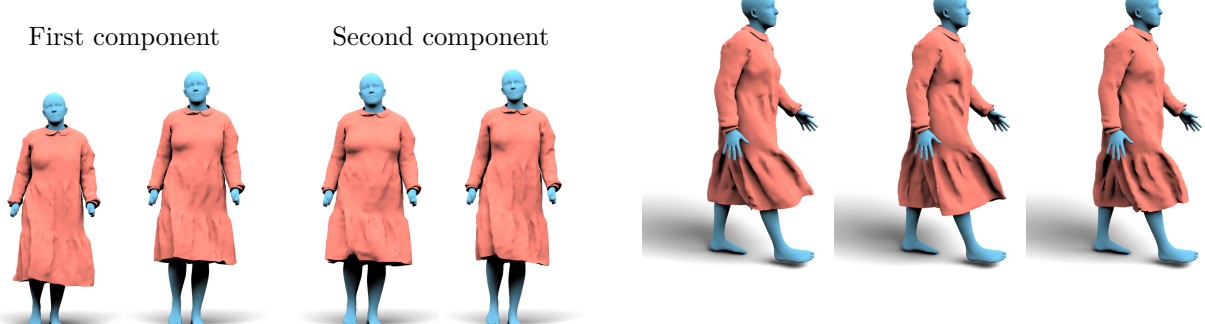

Figure 5: Results for variations of body shape. Our method adapts the garment to the body morphology.

Figure 6: For a given body motion and cloth material, our method can generate subtle yet plausible variations, by varying (from left to right) the random latent noise of the diffusion model.

et al., 2018), this method does not decouple the influence of multiple factors at once due to its limited capacity. For this reason, we compare to the three strong baselines MGDDG (Zhang et al., 2022), HOOD (Grigorev et al., 2023) and ContourCraft (Grigorev et al., 2024). MGDDG is trained on a sequence of 175 frames at 50 fps containing walking and running. The training uses the code provided by MGDDG. HOOD's training on diverse garments generalizes well to our dress without the need of fine-tuning its original pretrained model. ContourCraft is based on HOOD with additional losses to overcome self-collisions. To align timings across methods with varying framerates, we measure all metrics at 10 fps.

*D-Garment* outperforms all baselines across most metrics in Table 2, with HOOD and ContourCraft achieving comparable results only in terms of strain error $E_s$. Notably, our approach generalizes well to unseen body shapes and materials, though accuracy is slightly reduced for unseen motions. Given that the test set comprises examples with all factors unseen, our results demonstrate state-of-the-art performance for unseen generation tasks across different cloth material and body motion.

Figure 3 shows a visual comparison to HOOD, ContourCraft and MGDDG simulated on a body motion. The results of *D-Garment* are visually closer to the ground truth simulation. HOOD is struggling to handle pleats around the collar and the wrists because of its assumption of a flat rest shape. ContourCraft shows less intersections than HOOD but suffers from the same limitations. MGDDG's results are visually pleasing but do not follow the expected material behaviour. More visual results are provided in supplementary material.

## 5.3 Ablation Study

We perform ablations on the input parameters of the model, the mesh subdivision and the post-processing technique in Table 3. For material and motion ablations, we remove $\gamma$ and $\theta$ in the input vector of $\epsilon_\theta$, respectively, and train the full model. The motion ablation keeps the current pose but omits the prior ones

Table 3: Ablation study w.r.t. different conditioning and processing strategies. Best scores are in **bold**.

|  | Shape Similarity | | | Physical Validity | | | |
|---|---|---|---|---|---|---|---|
|  | $E_v$ | $E_{CD}$ | $E_n$ | $E_c$ | $E_b$ | $E_s$ | $E_d$ |
| w/o material | 7.0097 | 0.2421 | 0.4740 | **0.6677** | 0.6341 | 8.6457 | 3.5387 |
| w/o motion | 5.2396 | 0.1985 | 0.4576 | 0.7087 | 0.6013 | 8.4377 | 2.3849 |
| w/o post process | 4.9589 | 0.1888 | 0.4583 | 1.3632 | **0.5839** | 7.2277 | **2.3163** |
| w/ subdivision | - | **0.1854** | 0.4585 | 0.8145 | - | 11.1532 | 2.3414 |
| *D-Garment* | **4.9447** | 0.1863 | **0.4525** | 0.7180 | 0.5951 | **7.0201** | 2.3384 |

which indicate body velocity. The full model outperforms all ablations in most metrics. *D-Garment* can benefit from the post-processing reducing collision artifacts from 1.3% to 0.7% of intersecting vertices with a marginal physical quality trade-off. Variations in mesh resolution have a negligible impact on the deformed shape, demonstrating the robustness of the *uv* representation. Note that $E_v$ and $E_b$ are not applicable on meshes with different topology, while the physical validity metrics $E_c$ and $E_s$ are topology sensitive.

Next, we study the influence of material parameters and body shape in the *D-Garment* outputs. Figures 4 and 5 show results for extreme parameter values used during training. Figure 4 keeps all inputs constant while only varying a single cloth material parameter at a time. Note that each parameter influences the resulting geometry of the dress by subtly changing wrinkling patterns. This allows to generate a rich set of results by controlling cloth material. Figure 5 shows results when keeping all inputs constant while only changing a single parameter of the body shape $\boldsymbol{\beta}$, along the first two principal components. Note that the same dress is draped on different morphologies, while adapting its shape.

Figure 6 shows results obtained for the same conditions using different random noise for each latent. *D-Garment* allows for high variability, while staying in the space of plausible dynamics.

## 5.4 Garment fitting to 3D observations

We apply our method to fit temporally consistent garments on humans captured using multi-view videos, represented as sequence of unstructured 3D point clouds. This task presents challenges as the input data is noisy, and the points represent the human wearer in addition to the garment.

Using a recent method Toussaint et al. (2024), we first reconstruct point clouds of the multi-view dataset 4DHumanOutfit (Armando et al., 2023). We then fit a human motion prior (Marsot et al., 2022), modeled as a discrete set of SMPL parameters, to the reconstructed point clouds for each sequence. The body fitting process uses a one-sided Chamfer loss, distances to manually placed landmarks and the clothing energy $E_{cloth}$ from Yang et al. (2016) enforcing the body shape to stay inside the capture. Points in the capture are filtered for the cloth fitting process by removing points that are near undressed body parts (feet, legs, hands and head) and isolating the largest cluster of points. We test two material optimization strategies. The first uses fixed material parameters $\boldsymbol{\gamma}$ initialized to values with minimal constraints $(10^{-8}, 200, 0.01)$. The second strategy optimizes $\boldsymbol{\gamma}$ by optimizing $\boldsymbol{\gamma}^*$ in addition to $\mathbf{z}_T^*$ in Equation (6). We optimize for a single input latent of our model following Section 3.2 to fit generated garments to point cloud sequences.

Figure 7 shows a qualitative result for two of the sequences of 4DHumanOutfit (Armando et al., 2023). The input point cloud is shown in grey, and the fitted garment in orange. Visual results indicate that *D-Garment* can robustly recover the garment from dynamic 3D point clouds. Figure 8 shows quantitative results of the fitting application on these two sequences. We measure distances from vertices of *D-Garment* fittings to their nearest neighbors on reconstructed frames of the capture. All fitting results achieve an accuracy inferior to 2.4*cm* on 90% of garment vertices. A distance of 2.4*cm* between our result and the input point cloud is small compared to the highly dynamic garment motions present in 4DHumanOutfit. Furthermore, the error is overestimated for some parts as the dress model is complete, while the reconstructions in 4DHumanOutfit are missing parts due to occlusions in some frames, as shown in Appendix A, where the lower arm and parts of the torso are not reconstructed. Note that *D-Garment* fitting results achieve high accuracy despite

this domain gap. Optimizing for material parameters allows for smoother fitted results with a slight loss in surface to target distance.

## 6 Conclusion

This paper presented *D-Garment*, a template-specific 3D dynamic garment deformation model based on a 2D latent diffusion model enabling temporally consistent generation given the body shape, pose, motion and cloth material. The model was learned on a dataset of dresses with physical grounding by conditioning the result on cloth material parameters. Code and dataset are available at https://dumoulina.github.io/d-garment/. Experimental evaluations on both simulated and real data confirm the versatility of *D-Garment* for diverse tasks while providing state-of-the-art results. These findings highlight the potential of diffusion-based generative models to realistically deform 3D dynamic garments, as well as solving the challenging task of extracting garments from 3D point cloud observations.

## 7 Statement of Broader Impact

This research advances garment animation within creative industries, facilitating the realistic rendering of dressed avatars in diverse virtual environments. Furthermore, it holds significant potential for virtual try-on applications, aiming to mitigate high product return rates. However, the technology raises ethical concerns regarding potential misuse, particularly the generation of non-consensual realistic imagery and the associated privacy implications. Additionally, the reliance on substantial computational resources entails a negative environmental impact and limits accessibility for segments of the population.

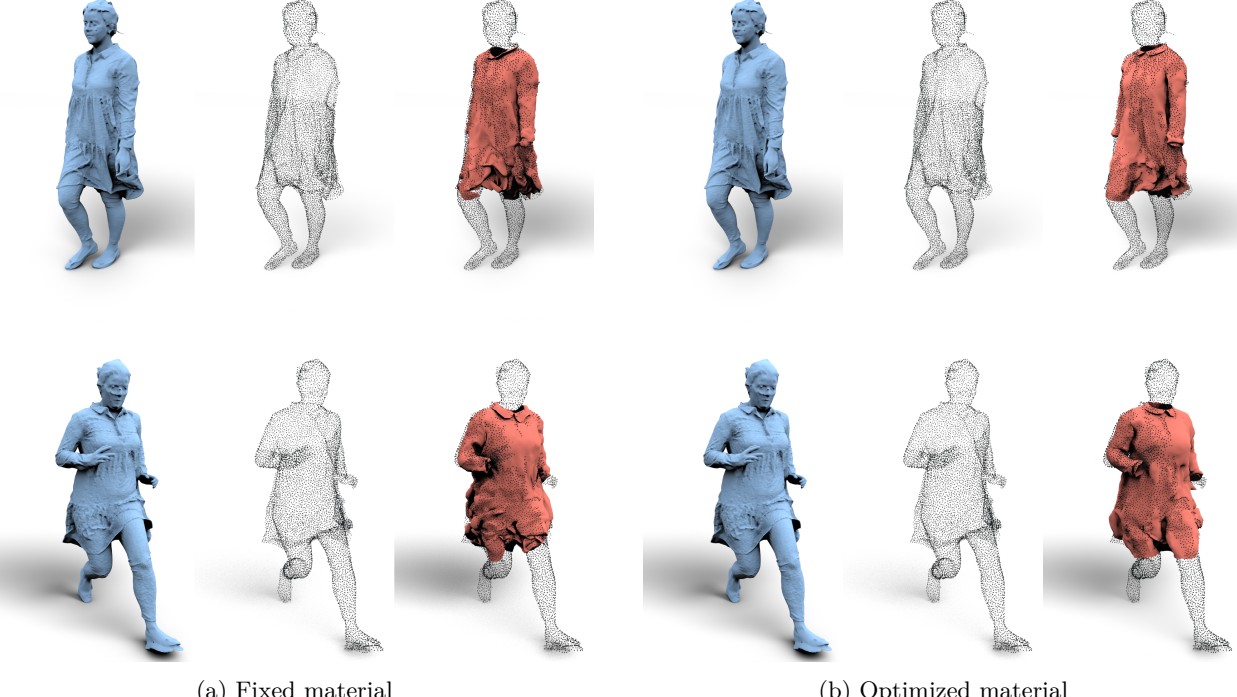

(a) Fixed material            (b) Optimized material

Figure 7: *D-Garment* is used to fit a captured 3D sequence. For each time frame of the corresponding motion (lines), we show the raw capture (left blue), the sampled point cloud (middle grey), and the estimated garment (right orange). We can accurately recover garment pose and fine wrinkles using two different material optimization strategies a and b. Corresponding videos are in supplementary material.

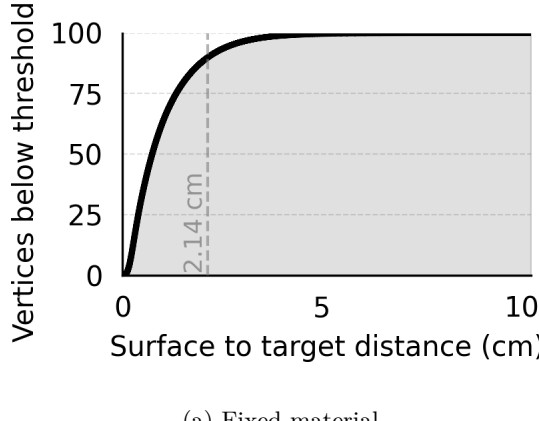 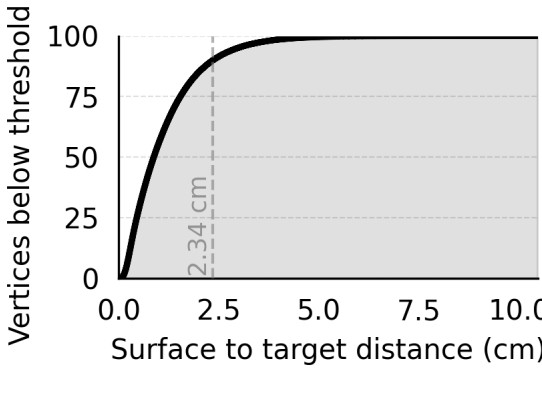

(a) Fixed material  (b) Optimized material

Figure 8: Cumulative distances of *D-Garment* fittings to input 3D point clouds for two sequences of 4DHumanOutfit with different material optimization strategies a and b. The vertical bar indicates the maximum distance at 90% of the vertices.

## Acknowledgments

This work was partially funded by the Nemo.AI laboratory by InterDigital and Inria. We thank João Regateiro and Abdelmouttaleb Dakri for helpful discussions, and Rim Rekik Dit Nkhili and David Bojanić for help with the SMPL fittings.

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

## A  Dress template and capture

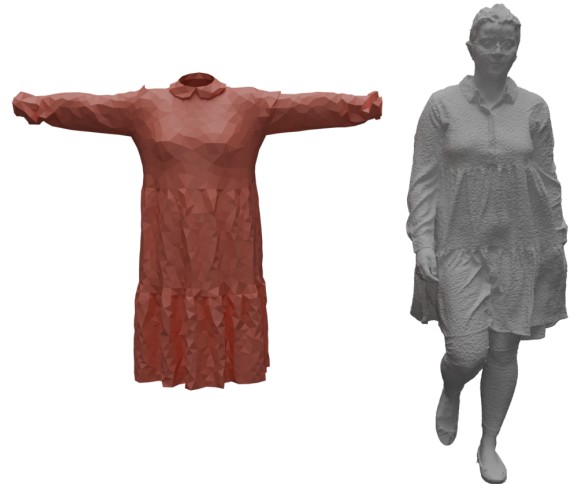

Figure 9: The dress model (left) used for *D-Garment* dataset differs slightly from the captured dress (right) from 4DHumanOutfit (Armando et al., 2023) in terms of length and wrinkle details.

## B  Garment generalization

*D-Garment* allows to realistically deform a fixed template. It therefore requires a separately trained model for each garment design. To show the model's applicability to different garments, we generated a dataset by simulating a T-shirt from Korosteleva et al. (2024) over a subsample of the original dataset described in Sec. 4. This dataset is significantly smaller than the original with 67 simulated sequences instead of 1519, resulting in 10089 training data points instead of 385448. We simulate one more test sequence with all input parameters unseen to provide unbiased results in Fig. 10 and Tab. 4. Fig. 10 shows that the new model can generate visually pleasing dynamic T-shirts. Quantitative results in Tab. 4 are similar to the results for the dress in the main paper, showing that *D-Garment* is applicable to different garments.

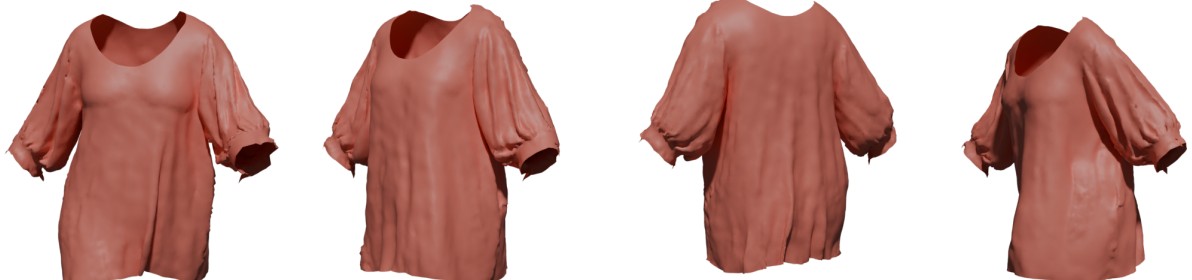

Figure 10: Visual results of *D-Garment* on 4 frames of a test sequence for the T-shirt model.

Table 4: Quantitative results of *D-Garment* trained for a T-shirt model using the test sequence.

|  | Shape Similarity | | | Physical Validity | | | |
|---|---|---|---|---|---|---|---|
|  | $E_v$ | $E_{CD}$ | $E_n$ | $E_c$ | $E_b$ | $E_s$ | $E_d$ |
| T-shirt | 3.59 | 0.05 | 0.42 | 0.03 | 0.46 | 1.21 | 2.13 |

## C   Influence of the pose sequence length

The pose sequence length $l$ defines the number of poses $\boldsymbol{\theta}$ input to the model. Tab. 5 shows quantitative results of different pose sequence lengths. Here, $l = 1$ corresponds to the static case, where no temporal information is available. The pose sequence length used throughout the paper is $l = 3$, which is the minimal length that allows to learn about dynamics caused by acceleration. We further show results for $l = 5$. *D-Garment* can benefit from a longer input sequence in terms of shape similarity metrics, while maintaining the same level of physical accuracy.

Table 5: Comparison of pose sequence length conditioning. Best scores are in **bold**.

| $l$ | Shape Similarity | | | Physical Validity | | | |
|---|---|---|---|---|---|---|---|
| | $E_v$ | $E_{CD}$ | $E_n$ | $E_c$ | $E_b$ | $E_s$ | $E_d$ |
| 1 | 5.2396 | 0.1985 | 0.4576 | **0.7087** | 0.6013 | 8.4377 | 2.3849 |
| 3 | 4.9447 | 0.1863 | 0.4525 | 0.7180 | **0.5951** | **7.0201** | 2.3384 |
| 5 | **4.8651** | **0.1811** | **0.4508** | 0.7450 | 0.5985 | 8.2247 | **2.2235** |

## D   Ablation of seam processing

The *uv* parametrization computed by Optcuts (Li et al., 2018) minimizes a combination of distortion, seam length, and the number of seams. However, remaining seams can cause discontinuities where multiple locations in the *uv*-map correspond to a single mesh vertex. To smooth boundaries, we average the values of all locations in the *uv*-map corresponding to the same vertex. We further interpolate the texture using bilinear filtering to smooth the texture globally. As demonstrated in Fig. 11, these techniques effectively preserve a coherent geometry.

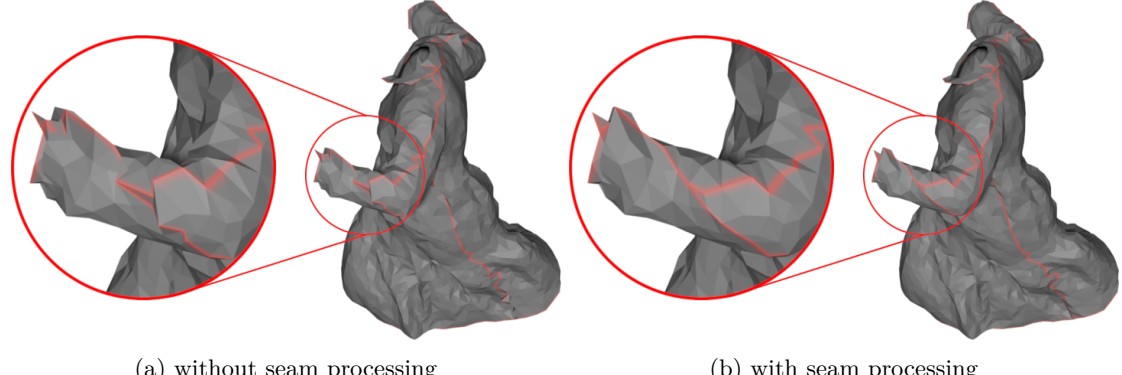

(a) without seam processing        (b) with seam processing

Figure 11: Visual comparison of the seam processing. *uv* seams are represented in red. Fig. 11a samples the nearest texel for each *uv*-coordinate and randomly picks between seam vertices, Fig. 11b averages seam vertices and interpolate the texture with bilinear filtering.

