# OpenReview forum: "D-Garment: Physically Grounded Latent Diffusion for Dynamic Garment Deformations"
_TMLR — Accepted by TMLR_

### Review · Reviewer_oXuo · 2026-01-25

**Summary Of Contributions:**

This paper presents D-Garment, a learning-based method for generating dynamic 3D garment deformations conditioned on body shape, motion, and physical cloth material properties. It encodes 3D mesh deformations as 2D displacement maps in UV space and uses geometry images approach to leverage powerful 2D diffusion models. It outperforms HOOD, ContourCraft, and MGDDG across all metrics (Table 2) and successfully fits to 4DHumanOutfit captures with <2.4cm error at 90% vertices.

**Audience:**

Yes

**Audience Explanation:**

To me using 2D latent diffusion for 3D mesh generation via UV mapping is an interesting design pattern that could transfer to other 3D domains.

Physical grounding: The conditioning on material parameters (rather than just pose) addresses a real limitation in prior work - interesting for physics-informed ML community.

**Broader Impact Concerns:**

Missing Broader Impact Statement:
The paper does NOT include a Broader Impact Statement, which is concerning given: Direct human-facing application (virtual try-on, avatars).  Socioeconomic implications (labor, accessibility). Use of human body data.

**Claims And Evidence:**

No

**Claims Explanation:**

Only tested on one dress type, generalization to pants, shirts, jackets unclear.
Dataset diversity is limited compared to methods like HOOD.
Test set has unseen motion + unseen shape + unseen material - extremely challenging but makes it hard to assess generalization on each factor independently.
Sequence length: Why only 3 poses (l=2)? Did you experiment with longer histories?
Material range: How did you select the uniform sampling ranges for γ? Are results sensitive to this?

**Requested Changes:**

Limited real-world testing, unclear generalization about dress type, motion, material range, lacking discussion of limitations and failure cases.

---

> ### Author Response · Authors · 2026-02-13
>
> # Dataset diversity is limited compared to methods like HOOD
>
> The diversity of our dataset can be discussed for each factor. *D-Garment* is trained for a fixed garment template. We simulate the variation of body shape and body material using uniformly sampled parameters, as in the case of HOOD (*cf.* Sec. 4). In our experiments, we limit our set of motions to running and walking motions, which occur most frequently in existing datasets such as AMASS.
>
> # Assessing generalization on each factor independently
>
> We included 3 new test set comparisons in Table 2 of the main paper, described in the 2nd paragraph of Sec. 5.1, with one unseen factor at a time showing that *D-Garment* is still competitive for each factor. Please refer to the 2nd paragraph of Sec. 5.2 for further analysis.
>
> # Sequence length
>
> The sequence length is a hyperparameter of our method. Appendix C was added to show the evolution of the quantitative error for different sequence lengths in Table 5. Note that setting the sequence length to 1 corresponds to the static case, where no temporal information is available. We chose a sequence length of 3 as this is the minimal length that allows to learn about dynamics caused by acceleration (which is the 2nd derivative of position). We now further evaluated the method for sequences of length 5 and found the method to be robust to longer pose sequences with a gain in terms of shape similarity metrics. Note that increasing the number of frames increases the computational and storage requirements of the method.
>
> # Material range
>
> We base our choice on the literature. We started from the extreme proposed values in Table 1 of [R1] and performed empirical testing so the simulation is stable. The goal was to have a broad range for generalization. For this reason, we set the range of the uniform sampling to the largest range that gives good solutions: $\mathbf{b} \sim \mathcal{U}([10^{-8}, 10^{-4}])$, $\mathbf{s} \sim \mathcal{U}([40, 200])$ and $\mathbf{d} \sim \mathcal{U}([0.01, 0.7])$ (*cf.* first paragraph of Sec. 4).
>
> [R1] Mickaël Ly, Jean Jouve, Laurence Boissieux, and Florence Bertails-Descoubes. Projective Dynamics with Dry Frictional Contact. *ACM Transactions on Graphics*, 39(4), 2020.

---

### Review · Reviewer_zDhU · 2026-02-01

**Summary Of Contributions:**

The authors present a method to generate 3D garment deformations by mapping them to 2D UV space and applying a Latent Diffusion Model. Unlike many existing draping methods, this approach explicitly conditions on physical parameters like stiffness and density, allowing for disentangled material control. To support this, they release a synthetic dataset of motion sequences created with Projective Dynamics. Finally, they show how the generative model can be used as a prior to fit noisy, real-world data from the 4DHumanOutfit dataset.

**Audience:**

Yes

**Audience Explanation:**

This work fits right into the current trend of mixing physics simulation with generative diffusion, which is obviously a hot area. It addresses some real-world bottlenecks in avatar creation, so I think it will be of interest to researchers working on neural rendering and geometric deep learning.

**Broader Impact Concerns:**

The paper focuses on garment simulation and reconstruction. No specific negative impact unique to this method stands out that would require a mandatory revision, but a standard statement on the responsible use of realistic avatar technology would be appropriate.

**Claims And Evidence:**

Yes

**Claims Explanation:**

The paper presents both geometric and physics-based evaluations. Comparison against HOOD and others in Table 2 looks good, with a fairly big drop in vertex error (4.9 compared to 19.7 for HOOD) and improved collision handling. Figures 4 and 5 demonstrate that the model actually disentangles material from shape. The application to real data in Figure 8 also shows decent results, fitting real scans within ~2cm. My main issue is that the generalization claim is somewhat overstate. It works on new motions/materials but seems limited to just one dress topology.

**Requested Changes:**

Critical adjustments:

1. My main issue is the scope of generalization. The method is presented as a general deformation model, but the authors only show results on a single dress topology. It wasn't clear if this works out-of-the-box for different garments or if it requires per-category retraining. If it's the latter, the claims in the abstract need to be toned down to reflect that it's template-specific.
2. Using 2D diffusion often leads to seam artifacts. I see the paper mentioned averaging to fix this, but visual proof is necessary. Can the authors show a close-up of the mesh boundaries? To make sure the "averaging" isn't just blurring out bad geometry?

Suggestions for improvement:

1. Regarding the fitting to 4DHumanOutfit, do the inferred physical parameters make sense? I'm curious if the optimized stiffness actually matches the visual material of the real garment, or if the model is just using those parameters to overfit the geometry. A short discussion here would strengthen the physical grounding argument.
2. The ablation in Table 3 shows a massive increase in strain error when using subdivision. This looks a bit weird. Is this due to high-frequency noise? An explanation of this behavior would be good.
3. Please compare the 7.5 fps speed against the baselines. Is it competitive? Also, is there a sweet spot for diffusion steps that balances speed and quality?

---

> ### Author Response · Authors · 2026-02-13
>
> # Question about seam artifacts
>
> The $uv$ seams create discontinuities in the 2D space which can cause artifacts on the 3D mesh if not handled correctly. We added Appendix D to show that our filtering technique successfully prevents such artifacts.
> In this appendix, Fig. 11 illustrates the effect of the seam processing.
>
> # Inferred physical parameters for 4DHumanOutfit
>
> The garments used in 4DHumanOutfit do not provide ground truth values for material parameters. Instead, the material is merely labeled by its material composition (97% polyester, 3% elastane). We can therefore not quantitatively measure the error of the estimated parameter values.
> Nevertheless, we noted that the fitted material falls within the range of the training distribution.
>
> # Strain error increases for subdivision
>
> For all metrics, we compare the result of *D-Garment* to the ground truth. At standard resolution, this ground truth is obtained using a simulator and can be considered accurate. For the subdivided mesh, we obtain the ground truth by subdividing the result simulated for lower resolution by simply adding vertices at the mid-points of all edges (thereby creating 4 coplanar triangles per triangle at lower resolution). The resulting output by *D-Garment* is smooth thanks to the bilinear interpolation in $uv$ space. This results in a strain metric $E_s$ that increases in this case.

---

### Review · Reviewer_asHP · 2026-02-03

**Summary Of Contributions:**

The paper proposes D-Garment, a method for dynamically deforming 3D garments using a conditional latent diffusion model. The core contributions are:
- A 2D Latent Diffusion Model (LDM) that takes conditional inputs: body shape, pose, pose velocity, and physical material of the garment and deforms the input 3D garment in the UV-space.
- Dataset Contribution:  A synthetic dataset of dynamic 3D dress with over 172 motions with variations of body shape and cloth materials.

**Additional Comments:**

**Strengths**
- The proposed method uses a 2D diffusion model instead of a 3D Diffusion model by employing uv-maps.
- The proposed method used body motion cues for garment deformation, which is a step forward in garment modeling.

**Audience:**

Yes

**Audience Explanation:**

This work is relevant for people working in graphics. Further, it has applications in the AR/VR industry (virtual try-on, gaming, etc.).

**Broader Impact Concerns:**

As this technology can be used to generate images with different clothing, I request that the authors add a statement regarding broader impact concerns.

**Claims And Evidence:**

Yes

**Claims Explanation:**

The authors claim that they present a physically grounded conditional generative model of 3D garment deformation. This claim is supported by the description of the method (Sec. 3) where the human-boyd motion is parametrized by the SMPL model and clothing material is inspired by physics-grounded simulation (Baraff & Witkin, 1998).

**Requested Changes:**

- **Comparison with missing baselines:** The authors did not compare with recent work Gaussian Garments [R1].
- **Generalization across garment types:** The authors report results for only a limited number of garment types. Does the method work on short jackets, long woolen cloths, loose trousers etc.? Is this a limitation of the method?
- Does the proposed method work for real-world images?
- Missing comparison of inference time with the baseline methods.


[R1] https://eth-ait.github.io/Gaussian-Garments/

---

> ### Author Response · Authors · 2026-02-13
>
> # Comparison to Gaussian Garments
>
> Gaussian Garment is a method that takes as input a multi-view video and outputs an animatable garment, defined as a simulation-ready representation that reconstructs the observed garment (geometry and appearance). The output is a mesh with a graph neural network that allows for neural simulation. The simulations shown in the Gaussian Garment paper are computed with ContourCraft. Note that only this last part is comparable to *D-Garment*, and the comparative evaluation with respect to ContourCraft is in our experiment in Sec. 5.2.
>
> # Does the method work on real-world images?
>
> *D-Garment* can currently be fitted to captured 3D data, as shown in Sec. 5.4. However, the method currently can not be directly applied to 2D images as *D-Garment* does not include an appearance model. In the future, *D-Garment* has the potential to be extended to image fitting applications by either learning an additional appearance model or *e.g.* by optimizing the garment with respect to vision foundational geometry network inferred depth or normal images.

---

### Author Response · Authors · 2026-02-13
**General Response to all Reviewers**

We thank all reviewers for their overall positive and constructive feedback. We appreciate their recognition of *D-Garment* employing a diffusion model in 2D space rather than in 3D (reviewer asHP), and *D-Garment* allowing for material control by conditioning on physical parameters (reviewer ZDhU), which is identified as interesting for the physics-informed ML community (reviewer oXuo).
We address below the reviewers’ comments. For compactness, questions raised by more than one reviewer are addressed here, while the remaining ones are answered in the individual responses. We believe the resulting experiments, clarifications, and discussions, presented in the **revision main and appendix (PDF)**, have made the manuscript more complete and thorough. All changes in the revised version of the paper are **highlighted in blue**.

# Applicability of the approach to other garment types

All reviewers raise the question of whether our approach can generalize to other garment types.
*D-Garment* learns how to deform a template based on input conditions corresponding to body shape, body motion, and cloth material, as illustrated in Fig. 2. *D-Garment* is a template-specific model that needs to be trained per template topology. The revised paper includes this specification in abstract, introduction, and conclusion. We also provide new results in Appendix B for a new T-shirt garment to show the applicability of our model to other garments after training on them.
In this appendix, Fig. 10 shows qualitative generation examples over 4 frames of a test sequence. We provide the error metrics of the T-shirt test set in Table 4 showing similar results to the original dress model.

# Statement of broader impact

All reviewers ask for a statement of broader impact, and we fully agree that this was missing. We added Sec. 7 to the paper.

# Comparison with respect to inference time

Reviewers asHP and zDhU ask questions about inference time / frame rate, in comparison to existing methods. We included average inference time in Table 2 of the main paper. *D-Garment* is on par with other methods. Diffusion is known for being slower, while allowing for better quality and less mode collapse than generative adversarial networks or variational auto-encoders. The diffusion process used in the submission can potentially be optimized with distillation techniques to reduce time steps.

---

### Decision · Action_Editor_R1MU · 2026-03-12

**Recommendation:** Accept as is

**Additional Comments:**

In their final recommendations, two reviewers proposed acceptance and one rejection, based on the main concern that the model needs to be trained for a specific template garment. While the AE agrees that this imposes constraints in terms of practical deployment, they believe that there remains value in the proposed work, as acknowledged by the other two reviewers.

**Audience:**

Yes

**Audience Explanation:**

All the reviewers acknowledged that there is a TMLR audience for this work.

**Claims And Evidence:**

Yes

**Claims Explanation:**

Initially, the reviewers expressed some concerns about the generality of the approach and its comparison to baselines. However, the authors' feedback addressed most of these concerns. Ultimately, all reviewers acknowledged that the claims made in the submission were sufficiently supported by clear evidence.